# Can an Exercise-Based Educational and Motivational Intervention be Durably Effective in Changing Compliance to Physical Activity and Anthropometric Risk in People with Type 2 Diabetes? A Follow-Up Study

**DOI:** 10.3390/ijerph16050701

**Published:** 2019-02-27

**Authors:** Francesca Gallè, Jesse C. Krakauer, Nir Y. Krakauer, Giuliana Valerio, Giorgio Liguori

**Affiliations:** 1Department of Movement Sciences and Wellbeing, University of Naples “Parthenope”, 80133 Naples, Italy; giuliana.valerio@uniparthenope.it (G.V.); giorgio.liguori@uniparthenope.it (G.L.); 2Metro Detroit Diabetes and Endocrinology, Southfield, MI 48034, USA; jckrakauer@gmail.com; 3Department of Civil Engineering, The City College of New York, New York, NY 10031, USA; nkrakauer@ccny.cuny.edu

**Keywords:** physical activity, anthropometric risk, type 2 diabetes

## Abstract

Aims. A nine-month motivational exercise-based intervention was previously offered to subjects with type 2 diabetes (T2D). A year after the end of the intervention, compliance to physical activity (PA) and anthropometric indices of participants were analyzed to evaluate the durability of its effects. Methods. PA levels, expressed as total energy expenditure per week, were assessed with the International Physical Activity Questionnaire (IPAQ). Changes in Body Mass Index (BMI), A Body Shape Index (ABSI), Hip Index (HI) z-scores, the relative mortality risk related to each of these measures, and a combined Anthropometric Risk Index (ARI) were also evaluated. Results. Of a total of the 52 subjects examined (67.9% males, mean age 61.8 ± 6.0), 46 (88.4%) were still sufficiently active as defined by IPAQ thresholds at follow-up. PA levels, anthropometric indices and related risks improved at follow-up in respect to the baseline and to the end of the intervention, although only PA levels, BMI and related measures, and ARI risk changed significantly. Habitual PA increased significantly after the intervention (*p* < 0.01) and this increase correlated with changes in BMI z-scores (*r* = −0.29, *p* = 0.04). BMI risk was significantly lower (*p* < 0.01) in participants still active at follow-up. Conclusions. This study testifies to the persistence of compliance to PA and health benefits of a combined exercise-based and motivational intervention in subjects with T2D.

## 1. Introduction

In the last three decades, the global prevalence of type 2 diabetes (T2D) has risen substantially, mirroring the increase of overweight and obesity [1]. In Italy, the prevalence rate of diabetes is 5.4% [2]. Regular physical activity (PA) may reduce the risk of developing T2D and contributes to weight and glycemic control in diabetic people, lowering their cardiovascular risk and improving their quality of life [1,3,4,5,6]. Current guidelines for individuals with T2D recommend at least 150 min/week of moderate-to-vigorous aerobic exercise over a period of 3 days per week, with no more than 2 consecutive days between bouts of aerobic activity, together with vigorous resistance training at least 2–3 days/week; moreover, individuals with T2D are encouraged to increase their total daily unstructured PA to improve their health status [7]. However, a large part of the world population does not meet the minimum recommended levels for PA per week, and the majority of subjects who are at highest risk for developing T2D or have diabetes do not engage in regular PA [1,8]. Therefore, public health interventions aimed to promote PA and increase exercise levels in these groups are needed. Multidisciplinary interventions including both exercise and behavioral/motivational counselling may be more effective than programs based exclusively on exercise sessions in addressing patient’s lifestyles [9,10,11,12,13,14].

During the years 2014–2015, a health promotion intervention for individuals with T2D funded by the National Center for Prevention and Control of Diseases of the Italian Ministry of Health was carried out in the city of Naples. It aimed to improve self-management of the disease and quality of life among these individuals by increasing their compliance to healthy lifestyles and behaviors through a nine-month exercise-based educational and motivational intervention [15]. The findings of that study showed a higher compliance to healthy diet and PA, better health outcomes, and enhanced physical fitness among participants to the intervention in respect to control subjects who underwent just an educational intervention [15,16,17].

The aim of the present study is to analyze the long-term efficacy of that intervention on behavior change and health risk of participants. Habitual PA levels and anthropometric values measured at a one-year follow-up were compared with those registered at the start and at the end of the intervention. In addition to BMI, we considered other allometric measurements for normalized waist circumference (A Body Shape Index, ABSI), normalized hip circumference (Hip Index, HI), and a combined Anthropometric Risk Index (ARI) [18,19,20] to quantify changes in mortality risk associated with the body size and shape of the participants.

## 2. Materials and Methods

This was a follow-up study evaluating PA levels and anthropometry-related health risk in a sample of individuals with T2D who took part in a nine-month supervised and combined exercise and motivational community-based intervention, one year after the end of the intervention [15]. The intervention was part of the community-based health promotion program funded by the National Center for Prevention and Control of Diseases of the Italian Ministry of Health. 

### 2.1. Participants and Setting

One year after the end of the intervention, participants (aged between 50 and 70 years, and presenting T2D without major complications) were invited to take part in the follow-up collection of data. The development of severe limitations to PA and the participation to other exercise or motivational programs throughout the year were considered as exclusion criteria. All participants were informed about the purpose of the study and the use of resulting data, and provided written informed consent.

### 2.2. Intervention

The activities included in the intervention have been previously described [15,16,17]. In brief, sedentary subjects with T2D were invited by their physicians to participate to the study. The intervention lasted nine months and included a motivational and educational program focused on PA and diet behaviors, and exercise sessions performed two times per week. The motivational program was structured in bi-weekly group sessions and focused on diet and PA benefits, suggestions of home-based and outdoor training methods, analysis of barriers, and problem solving in diabetes management. The nutritional program included quarterly group meetings aiming to highlight the benefits of a Mediterranean diet, healthy food choices and the adequate daily distribution of meals in diabetes control. The exercise program consisted of one-hour group sessions of adapted physical activity performed two times per week on non-consecutive days. It included moderate-to-vigorous aerobic and resistance exercises. All the activities were free of charge and were carried out in accordance with the Italian ethical standards and with the Helsinki Declaration.

### 2.3. Outcomes

#### 2.3.1. Behavioral Outcomes

Habitual Physical Activity was assessed through the short format of the International Physical Activity Questionnaire (IPAQ) [21], which assesses the total energy expenditure per week by considering minutes spent on vigorous/moderate-intensity (8/4 Metabolic Equivalents - METs) PA and walking (3.3 METs). The IPAQ total score is expressed in MET-minutes/week and represents an index of inactivity (<600 MET-minutes/week) or minimal/high activity (≥600 MET-minutes/week). The IPAQ was administered by the same trained researchers. 

#### 2.3.2. Anthropometric Outcomes

Anthropometric measurements were assessed by physicians. BMI was obtained by measuring height and weight using a medical-certified scale and a stadiometer. Waist and hip circumference (WC, HC) were assessed using a non-stretchable tape and expressed in centimeters to the nearest 0.1 cm. WC was measured at the end of a normal expiration between the lowest border of the rib cage and the upper border of iliac crest; HC was measured at the widest part of the hip, at the level of the greater trochanter [22].

#### 2.3.3. Statistical Analyses

A descriptive analysis was carried out to highlight possible variations of the considered outcomes from the start (T_0_) to the end of the intervention (T_1_), and at follow-up (T_2_). Habitual PA levels and time spent in PA per week, BMI, WC, and HC were expressed as mean values ± standard deviation (SD). ABSI and HI were determined using previously published formulas based on power-law relationships found between height, weight, WC, and HC [17,18,19]. BMI, ABSI, and HI values were converted to age- and sex-specific z-scores. The relative mortality risk based on BMI, ABSI, and HI was then calculated. The BMI, ABSI, and HI population normal values and associated mortality hazard ratio (HR) functions used for these calculations were taken from a previously published analysis of a national population sample in the United States with an approximately 20-year follow-up for mortality [19]. Each mortality HR function was expressed relative to the United States population mean, so that 1 represents the average hazard [20]. ARI is the product of the HRs attributable to each individual’s BMI, ABSI, and HI values, and represents a fuller measure of risk based on individual anthropometric profile and the approximate statistical independence of the component indices [20]. A one-way repeated measures ANOVA was performed to analyze the overall changes throughout the observation period; pairwise comparisons between groups were done with the Bonferroni post-hoc test. The sample size was calculated by considering, as the main outcome, the increase of METs-min/week needed to reach the minimal/high activity level respect to baseline. A Pearson’s correlation analysis was carried out to explore the association between changes in the anthropometric indices and those in weekly PA registered at one-year follow-up with respect to the end of the intervention. 

A *p*-value of 0.05 was assumed as the significance threshold. Data were analyzed with IBM SPSS Statistics for Windows, version 24 (Armonk, NY, USA; IBM Corp.).

## 3. Results

Fifty-two out of the 69 subjects (67.9% males, mean age 61.8 ± 6.0, range 44–75 years) who participated in the exercise-based educational and motivational intervention agreed to take part to the follow-up assessment (75.4% participation rate). This sample size gave a statistical power higher than 80%. Forty-six subjects (88.4%) were still sufficiently active as defined by the IPAQ thresholds (≥600 METs-min/week) a year after the end of the intervention (Figure 1).

Table 1 shows the mean characteristics of the 52 subjects at the start (T_0_) and end (T_1_) of the intervention, and at one-year follow-up (T_2_). Energy expenditure per week, so as minutes spent in total, vigorous and moderate PA, were higher at the end of the intervention and at the one-year follow-up compared to T_0_. Regarding the anthropometric variables, an overall improvement was found at the end of the intervention and also at the follow-up; the decrease of weight, waist circumference, BMI, and related z-score were significant in respect to T_0_, while the other outcomes did not change significantly. Variations registered after the end of the intervention were not significant, with the only exception of weekly energy expenditure. 

Table 2 shows the average hazard ratios associated with BMI, ABSI, HI, and ARI in participants. All these values decreased at the end of the intervention and at follow-up. Reductions registered in respect to baseline values were significant for BMI and ARI risk; no significant differences were registered between post-intervention and follow-up times.

Table 3 reports the results of Pearson’s correlation carried out between the changes of anthropometric indices and those of weekly energy expenditure (expressed in MET-minutes/week) between the end of the intervention and the one-year follow-up. METs variations were significantly related only to BMI z-score variations; a significant correlation was also found between changes in ABSI and HI z-scores.

## 4. Discussion

The findings of this study demonstrate the efficacy of a nine-month exercise-based educational and motivational intervention in determining a durable behavior change and an improvement of anthropometric measures in people with T2D. In fact, habitual PA levels of participants showed a significant increase from pre- to post-intervention and also at follow-up. In particular, the time spent in moderate-to-vigorous physical activity showed an increase at the end of the intervention in relation to the exercise program delivered, and a subsequent non-statistically significant decrease at follow-up, while the total amount of time spent in PA furtherly increased even after the end of the intervention, which testifies to the general adoption of an active lifestyle including low-intensity activities. Furthermore, it has to be noted that the majority of the sample was still active at follow-up, despite no other organized activities being proposed to the participants during the year after the intervention: therefore, it is possible that the reported increase in PA levels may be ascribed to the previous motivational intervention, which was able to improve awareness about the beneficial effects of PA and self-efficacy in the participants. The change in PA observed in our sample is in line with the findings of De Greef et al. in Belgian type 2 diabetic patients attending a cognitive-behavioral intervention [9]. However, contrary to our experience, the Belgian study reported lower levels of PA at follow-up than at the end of the intervention: this suggests that a combined intervention including motivational and exercise paths may be more effective than a motivational-only intervention to determine behavior changes in subjects with T2D.

In our sample, the increase of habitual PA was accompanied by a significant reduction in mean weight, BMI, BMI z-score, and related risk; furthermore, changes in PA levels at follow-up showed a correlation with BMI z-score. The other anthropometric parameters improved at the end of the activities and also at follow-up, even if not significantly, confirming the efficacy and the durability of the intervention.

In addition to BMI, we also explored other anthropometric measures to analyze health risk and its variations in the participants. ABSI and HI were derived to be indicators for WC and HC independent of BMI [17,18,19]. A series of studies from different countries have found ABSI to be positively associated with total and cardiovascular mortality [23,24]. Limited experience with HI shows a U-shaped relationship to mortality similar to BMI [20]. The anthropometric risk index (ARI) is calculated by combining the hazard ratios due to each of these independent indicators, and represents a strong linear predictor of mortality hazard [20]. Therefore, it could provide a reliable combined risk estimate. Initially tested on the US adult population to assess mortality risk, these indices have been extended to other populations and to other outcomes, such as cardiovascular and metabolic ones, to produce risk estimates with the promise to help in medical decision making [23,24,25,26,27,28,29,30,31,32,33,34,35,36]. Therefore, we decided to employ these indices in our analysis to evaluate the possible effects of the intervention on the mortality risk profiles of participants. 

In the present study, all these anthropometric variables and related indices showed improvements at the end of the intervention and at follow-up compared to baseline, even if these changes were not all significant. Furthermore, the increase registered in PA levels during the follow-up year was correlated with the improvement in BMI z-score. These findings confirm the role of PA in improving risk profiles through a modification of anthropometric features, and they speak to the usefulness of the proposed intervention. A longer follow-up, evaluating further anthropometric improvements, may probably show more consistent variations in the considered parameters.

This study has some limitations. First of all, the sample size was small and the involvement of the only participants in the previous intervention may have produced a selection bias. The follow-up analysis of behaviors and physical parameters among those subjects who participated in the previous study as controls might have given stronger findings, but they were not compliant. In addition, since the program was successful in improving PA levels in the majority of the sample, it was not possible to compare risk indices between active and sedentary individuals at follow-up. Indeed, this last group was too small to perform statistical analyses.

Furthermore, we did not analyze the possible changes in the nutritional behaviors of participants, which could have contributed to the improvement of anthropometric parameters.

Moreover, this study was based on the measurement of habitual PA levels through a questionnaire. Although the IPAQ represents a standard measure tool to assess energy expenditure, it provides self-reported information, which could be unreliable; therefore, the use of an objective instrument such as an accelerometer could have made the study more accurate.

Finally, in this study anthropometric risk indices were normalized on the reference values derived from a US population, assuming that the eventual differences are not considerable, but this might have affected the results.

Because of these limitations, further research is needed to confirm the benefits seen in other populations. Nevertheless, our findings suggest that similar integrated interventions may produce durable, positive lifestyle changes in subjects with T2D, leading to better health outcomes related to anthropometric features. Notwithstanding all its limitations, the study analyzed the durability of the effects of a lifestyle program and introduced novel anthropometric indices besides the traditional ones. These aspects represent a novelty in the field and attribute significance to the study.

## 5. Conclusions

The results of this study support the efficacy of an educational and motivational exercise-based intervention in improving compliance to PA, anthropometric measures, and related risk in subjects with T2D. In particular, participants reported higher levels of habitual PA at the end of the intervention and also at follow-up, and this seems to be associated with a reduction in mortality risk as predicted by the anthropometric indices. These findings should be considered in order to improve the self-management of T2D for public health purposes. 

## Figures and Tables

**Figure 1 ijerph-16-00701-f001:**
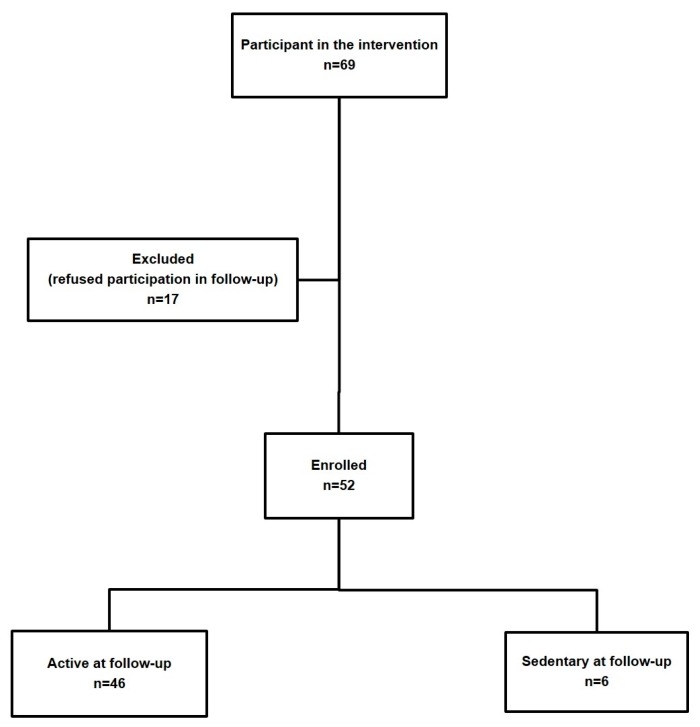
Flowchart for enrollment of participants in the follow-up study and their classification on the basis of habitual PA levels.

**Table 1 ijerph-16-00701-t001:** PA levels and anthropometric measures from the start of the intervention to the follow-up.

Outcome	T_0_	T_1_	T_2_	T_0_–T_1_ *p*	T_1_–T_2_ *p*	T_0_–T_2_ *p*	ANOVA *p*
Habitual PA MET-min/week	474.8 ± 90.3	641 ± 53.5	663.4 ± 57.0	<0.01	1	<0.01	<0.01
minutes/week spent in PA	122 ± 35.2	153 ± 10.7	168 ± 25.3	<0.01	<0.01	<0.01	<0.01
vigorous PA	10 ± 5.5	20 ± 5.2	15 ± 5.6	<0.01	0.06	<0.01	<0.01
moderate PA	36 ± 6.3	60 ± 4.5	55 ± 10.5	<0.01	0.13	<0.01	<0.01
Weight Kg	83.7 ± 8.8	79.1 ± 8.5	76.4 ± 9.2	0.03	0.38	<0.01	<0.01
Waist circumference cm	99.5 ± 6.3	93.9 ± 7.9	91.3 ± 9.2	<0.01	0.32	<0.01	<0.01
Hip circumference cm	97.8 ± 6.8	96.5 ± 7.2	95.5 ± 8.5	1	1	0.41	0.32
BMI Kg/m^2^	30.0 ± 2.6	28.3 ± 2.4	27.4 ± 2.7	<0.01	0.17	<0.01	<0.01
BMI z-score	0.50 ± 0.51	0.17 ± 0.46	−0.01 ± 0.51	<0.01	0.16	<0.01	<0.01
ABSI (m^11/6^ Kg^−2/3^)	0.08 ± 0.004	0.07 ± 0.005	0.07 ± 0.005	0.29	1	0.14	0.10
ABSI z-score	−0.83 ± 0.99	−1.2 ± 1.2	−1.3 ± 1.4	0.27	0.24	0.14	1
HI cm	91.8 ± 5.1	93.1 ± 5.7	93.7 ± 6.6	0.73	1	0.27	0.21
HI z-score	−1.8 ± 1.3	−1.4 ± 1.5	−1.3 ± 1.7	0.68	1	0.23	0.19

Mean values (±SD) of habitual physical activity (PA), time spent in PA and anthropometric parameters of participants (n = 52) measured before (T_0_) and at the end (T_1_) of the intervention, and at follow-up (T_2_), with *p* values from ANOVA and Bonferroni post hoc test pairwise and overall comparisons.

**Table 2 ijerph-16-00701-t002:** Anthropometric risk measures from the start of the intervention to follow-up.

Outcome	Mean Hazard Ratios			T_0_–T_1_ *p*	T_1_–T_2_ *p*	T_0_–T_2_ *p*	ANOVA
	T_0_	T_1_	T_2_				*p*
BMI risk	1.00 ± 0.09	0.94 ± 0.08	0.92 ± 0.08	<0.01	0.43	<0.01	<0.01
ABSI risk	0.90 ± 0.09	0.88 ± 0.07	0.87 ± 0.09	0.89	1	1	0.58
HI risk	1.16 ± 0.16	1.12 ± 0.16	1.11 ± 0.16	0.66	1	0.39	0.27
ARI risk	1.04 ± 0.14	0.93 ± 0.13	0.91 ± 0.13	<0.01	1	<0.01	<0.01

Risk values (estimated hazard ratios for mortality relative to a population mean) calculated for BMI, ABSI, HI, and ARI before (T_0_), at the end (T_1_) and at follow-up (T_2_) of the intervention, with corresponding *p* values from ANOVA and Bonferroni post hoc test pairwise and overall comparisons.

**Table 3 ijerph-16-00701-t003:** Correlation among variations registered at follow-up in anthropometric indices and PA levels.

T_1_–T_2_ Changes	Δ BMI z-Score *p*	Δ ABSI z-Score *p*	Δ HI z-Score *p*	Δ PA *p*
Δ BMI z-score *p*	1	−0.20 0.15	−0.07 0.61	−0.29 0.04
Δ ABSI z-score *p*	−0.20 0.15	1	−0.74 <0.01	0.15 0.27
Δ HI z-score *p*	−0.07 0.61	−0.74 <0.01	1	0.08 0.56
Δ PA *p*	−0.29 0.04	−0.15 0.27	0.08 0.56	1

Results (*r* coefficients and *p* values) from Pearson’s correlation among changes in anthropometric indices z-scores and in weekly energy expenditure (Δ PA) registered between the end of the intervention (T_1_) and follow-up (T_2_).

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
