# Peer review of "Can an Exercise-Based Educational and Motivational Intervention be Durably Effective in Changing Compliance to Physical Activity and Anthropometric Risk in People with Type 2 Diabetes? A Follow-Up Study"

_ijerph, 2019, doi:10.3390/ijerph16050701_

Round 1

Reviewer 1 Report

Major concerns

The authors did provide enough details (or content) of the 9 month exercise-based educational and motivational intervention programme. 

Line 69. How did the authors ensure that the participants were not involved in any other form of exercise or motivational progarmme since they were informed of this part of the study only at the end of this study period of 1 year?

Statistical analyses. Authors used Students paired and unpaired t-tests to analyze the results. Reviewer think this is inappropriate. A 2-way ANOVA comparing between 2 groups (experimental and control) with 3 time points (T0, T1 and T2), should be used.

Minor concern

- Line 71. "The interval developmnet of several limitations to ....". This sentence is not clear.

- Figures 1-4 are not clear. Need to explain what does numerical value of the level of risk in the Y-axis means to the reader. 

Author Response

Reviewer 1

Major concerns

The authors did provide enough details (or content) of the 9 month exercise-based educational and motivational intervention programme. 

As requested by Reviewer 2, we have added other details of the intervention (line 78):

“The motivational program was structured in bi-weekly group sessions and focused on diet and PA benefits, suggestions of home-based and outdoor training methods, analysis of barriers and problem solving in diabetes management. The nutritional program included quarterly group meetings aimed to highlight the benefits of Mediterranean diet, healthy food choices and adequate daily distribution of meals in diabetes control. The exercise program consisted of one-hour group sessions of adapted physical activity performed two times per week on non-consecutive days. It included moderate-to-vigorous aerobic and resistance exercises.”

Line 69. How did the authors ensure that the participants were not involved in any other form of exercise or motivational progarmme since they were informed of this part of the study only at the end of this study period of 1 year?

The participation to other motivational or exercise programs was considered an exclusion criterion. The sentence has been rephrased as follows (line 70):

“The intercurrent development of severe limitations to PA and the participation to other exercise or motivational programs throughout the year were considered as exclusion criteria.”

Statistical analyses. Authors used Students paired and unpaired t-tests to analyze the results. Reviewer think this is inappropriate. A 2-way ANOVA comparing between 2 groups (experimental and control) with 3 time points (T0, T1 and T2), should be used.

A control group was not included in the present study, due to the fact that only participants to the intervention agreed to participate to follow-up. We apologize for the misleading due to the citation of the control group included in the previous study in the first version of the manuscript. Now we have removed any reference to control patients. A repeated measures 1-way ANOVA and the Bonferroni post-hoc test have been performed to compare the results found at the three collection times. Results description has been changed on the basis of these new analyses.

Minor concern

- Line 71. "The interval developmnet of several limitations to ....". This sentence is not clear.

The term “interval” was wrong; it has been changed with “intercurrent” and the sentence has been changed as reported above.

- Figures 1-4 are not clear. Need to explain what does numerical value of the level of risk in the Y-axis means to the reader. 

The description of the figures have been changed as follows:

“Figs 2-5 show the distribution of BMI risk, ABSI risk, HI risk and ARI risk values between the participants (n=6) who were sedentary at follow-up (METs cod=0) and those who were still active (n=46, METs cod=1). Those who maintained sufficient PA levels after a year since the end of the intervention had a significantly better BMI risk profile than their inactive counterpart (p<0.01), while no significant difference was found regarding the other anthropometric risk profile components (ABSI risk p=0.65, HI risk p=0.48, ARI risk p=0.10).”

Thank you for these suggestions that have helped us to improve the manuscript.

Reviewer 2 Report

This study is of interest to the field but it needs a clearly defined research question, additional information on the sample and a better discussion structured according to the research questions. I therefore recommend a major revision. Please see my comments below.

Since this study was design as a retrospective study, it is difficult to conclude effectiveness of the intervention, such as "increase (L182-184)" and "high level (L211)". 

-L36 Reference is needed.

-L65 Flowchart would make this section more easy to understand. It is not clear how to recruits 222 patients (69+90, what about others?)

-L122 Male?

Author Response

Reviewer 2

This study is of interest to the field but it needs a clearly defined research question, additional information on the sample and a better discussion structured according to the research questions. I therefore recommend a major revision. Please see my comments below.

The authors are grateful to the Reviewer 2 for his valuable suggestions.

- Since this study was design as a retrospective study, it is difficult to conclude effectiveness of the intervention, such as "increase (L182-184)" and "high level (L211)". 

As suggested by another Reviewer, we have removed the term “retrospective” throughout the text and considered this as a follow-up study focused on specific items resulting by an intervention aimed at increasing compliance to physical activity and quality of life in people with type 2 diabetes.

-L36 Reference is needed.

Considering that both sentences refer to the same guidelines, we have linked them as follows (line 35):

“Current guidelines for individuals with T2D recommend at least 150 min/week of moderate to vigorous aerobic exercise over a period of 3 days per week, with no more than 2 consecutive days between bouts of aerobic activity, together with vigorous resistance training at least 2–3 days/week; moreover, individuals with T2D are encouraged to increase their total daily unstructured PA to improve their health status [7].”

-L65 Flowchart would make this section more easy to understand. It is not clear how to recruits 222 patients (69+90, what about others?)

The number 222 referred to the patients recruited for the controlled intervention. Since the present study was aimed to analyze the durability of the effects of the intervention among the only 69 participants, we have removed any reference to the previous control group (90 subjects). Anyway, we have added a flowchart (Figure 1) to make the study design easier to understand.

-L122 Male?

Yes, the letter M indicated males; it has been changed into the word “males”.

Reviewer 3 Report

Review ijerph-409332 – “Can an exercise-based educational and motivational intervention be durably effective in changing compliance to physical activity and anthropometric risk in people with type 2 diabetes?”

The manuscript presents 1-year follow-up anthropometric data following a nine-month exercise intervention (that included some educational and motivational components) in people with type 2 diabetes.  While the study may have been well designed, the design is not appropriately explained in the manuscript and the appropriate statistics are not used to test the study questions with this study design.  I have made a variety of major and minor comments to help make this study more clear.

Major Comments.

The control group needs to be described in the methods and included in the results.  The authors will need to redo all of the statistics/results and include the control group.  Paired t-tests are not appropriate for this study design.  The authors should consider repeated measures ANOVAs or other types of analysis that can compare group effect, time effect and group x time interactions.

This is not a retrospective study, rather this study is an intervention, but the researchers are focusing on the 1-year follow up data.  This study design needs to be properly explained so that readers can understand can more easily follow the manuscript.

Minor Comments.

Abstract: 

1.     Line 19: I am assuming M = Men, but please be more clear.

2.     Line 20: What does active mean?  Please define what you are considering physically active.

3.     How long was the initial intervention?  (Reading further I see 9months).  The study design needs to be clear in the abstract.

Materials and Methods:

1.     Line 71:  What does “interval” development of severe limitations mean?  What were the other exclusion criteria for the initial study?  Were the study participants sedentary before the start of the study?

2.     Line 72: “All actives were free of charge.”  Where there activities available to the participants after the intervention?

3.     Line 76:  While I respect that you do not need to go into detail about the initial intervention, the reader should know the basic goals, i.e. how much physical activity were you trying to elicit?  Was it weight training or cardiovascular activities?  A mix?  Also summarize the motivational intervention and educational program briefly.

4.     Line 79: Please describe the control group.  Were the participants randomized into the intervention and control group?

5.     Line 111:  The statistics seem inappropriate for the study design.

Results

1.     Table 1.  Total physical activity minutes/week would also be helpful.  It would also be helpful to know the amount of time spent doing moderate physical activity and the amount of time spent doing vigorous physical activity.

Author Response

Reviewer 3

Review ijerph-409332 – “Can an exercise-based educational and motivational intervention be durably effective in changing compliance to physical activity and anthropometric risk in people with type 2 diabetes?”

The manuscript presents 1-year follow-up anthropometric data following a nine-month exercise intervention (that included some educational and motivational components) in people with type 2 diabetes.  While the study may have been well designed, the design is not appropriately explained in the manuscript and the appropriate statistics are not used to test the study questions with this study design.  I have made a variety of major and minor comments to help make this study more clear.

The authors are grateful to the Reviewer 3 for his valuable suggestions.

Major Comments.

The control group needs to be described in the methods and included in the results.  The authors will need to redo all of the statistics/results and include the control group.  Paired t-tests are not appropriate for this study design.  The authors should consider repeated measures ANOVAs or other types of analysis that can compare group effect, time effect and group x time interactions.

We apologize for the complexity of the previous text: actually, it did not allow to easily understand the design of the study. In the present study a control group was not included: only those subjects who participated to the intervention accepted to take part to the subsequent follow-up, and the manuscript was focused on them. In the previous version of the manuscript we cited the control group of the intervention, but this was clearly misleading for Reviewers and may be misleading for readers. Now we have deleted any reference to the intervention controls. A repeated measures 1-way ANOVA and the Bonferroni post-hoc test have been performed to compare the results found at the three collection times. Results description has been changed on the basis of these new analyses.

This is not a retrospective study, rather this study is an intervention, but the researchers are focusing on the 1-year follow up data.  This study design needs to be properly explained so that readers can understand can more easily follow the manuscript.

We have removed the term “retrospective” throughout the text and we have tried to make the study design more clear. Thank you for your suggestions.

Minor Comments.

 Abstract: 

1.        Line 19: I am assuming M = Men, but please be more clear.

Yes, the letter M indicated males; it has been changed into the word “males”.

2.    Line 20: What does active mean?  Please define what you are considering physically active.

The sentences regarding PA levels in methods and results sections have been changed as follows (line 16):

“PA levels, expressed as total energy expenditure per week, were assessed with the International Physical Activity Questionnaire (IPAQ).”…. “On a total of 52 subjects examined (69.2% male, mean age 63±6.03), 46 (88.4%) were still sufficiently active as defined by IPAQ thresholds at follow-up.”

3.    How long was the initial intervention?  (Reading further I see 9months).  The study design needs to be clear in the abstract.

The aims section has been changed as follows (line 13):

“A nine-month motivational exercise-based intervention was previously offered to subjects with type 2 diabetes (T2D). After a year since the end of the intervention, compliance to Physical Activity (PA) and anthropometric indices of participants were analyzed to evaluate the durability of its effects.”

Materials and Methods:

1.        Line 71:  What does “interval” development of severe limitations mean?  What were the other exclusion criteria for the initial study?  Were the study participants sedentary before the start of the study?

The correspondent sentences have been changed as follows (line 68):

“After one year since the end of the intervention, participants (aged between 50-70 years and presenting T2D without major complications) were invited to take part to the follow-up collection of data. The intercurrent development of severe limitations to PA and the participation to other exercise or motivational programs throughout the year were considered as exclusion criteria.”

Sedentary patients represented the original target of the intervention. We have changed the term “inactive” with “sedentary”.

2.        Line 72: “All actives were free of charge.”  Where there activities available to the participants after the intervention?

No, the sentence was referred to the activities included in the motivational exercise-based program and is included in the description of the previous intervention.

3.        Line 76:  While I respect that you do not need to go into detail about the initial intervention, the reader should know the basic goals, i.e. how much physical activity were you trying to elicit?  Was it weight training or cardiovascular activities?  A mix?  Also summarize the motivational intervention and educational program briefly.

More details of the programs have been added (line 78):

“The motivational program was structured in bi-weekly group sessions and focused on diet and PA benefits, suggestions of home-based and outdoor training methods, analysis of barriers and problem solving in diabetes management. The nutritional program included quarterly group meetings aimed to highlight the benefits of Mediterranean diet, healthy food choices and adequate daily distribution of meals in diabetes control. The exercise program consisted of one-hour group sessions of adapted physical activity performed two times per week on non-consecutive days. It included moderate-to-vigorous aerobic and resistance exercises.”

4.     Line 79: Please describe the control group.  Were the participants randomized into the intervention and control group?

In the present study a control group was not included (see above). Now we have deleted any reference to the intervention controls and we hope that the design of the study could be more clear.

5.     Line 111:  The statistics seem inappropriate for the study design.

A repeated measures 1-way ANOVA and the Bonferroni post-hoc test have been performed to compare the results found at the three collection times. Results description has been changed on the basis of these new analyses.

Results

1.     Table 1.  Total physical activity minutes/week would also be helpful.  It would also be helpful to know the amount of time spent doing moderate physical activity and the amount of time spent doing vigorous physical activity.

These information have been added to the Table 1, described in the results and commented in the discussion as follows (lines 136 and 194):

“Energy expenditure per week, so as minutes spent in total, vigorous and moderate PA, increased at the end of the intervention and at the one-year follow-up compared to T0.”

“In particular, although the time spent in moderate-to-vigorous activity, which increased during the intervention in relation to the exercise program delivered, decreased during the follow-up year, the total amount of time spent in PA furtherly increased during the same period, testifying the general adoption of an active lifestyle including low-intensity activities.”

Round 2

Reviewer 1 Report

None

Author Response

We are grateful to Reviewer 1 for his positive consideration.

Reviewer 3 Report

The manuscript is improved.  I do not have any other comments.  

Author Response

We are grateful to Reviewer 3 for his positive consideration.